# A Simple Approach for State-Action Abstraction using a Learned MDP Homomorphism

## Abstract

Animals are able to rapidly infer, from limited experience, when sets of state-action pairs have equivalent reward and transition dynamics. On the other hand, modern reinforcement learning systems must painstakingly learn through trial and error that sets of state-action pairs are value equivalent—requiring an often prohibitively large amount of samples from their environment. MDP homomorphisms have been proposed that reduce the observed MDP of an environment to an abstract MDP, which can enable more sample efficient policy learning. Consequently, impressive improvements in sample efficiency have been achieved when a suitable MDP homomorphism can be constructed *a priori*—usually by exploiting a practitioner's knowledge of environment symmetries. We propose a novel approach to constructing a homomorphism in discrete action spaces, which uses a partial model of environment dynamics to infer which state-action pairs lead to the same state—reducing the size of the state-action space by a factor equal to the cardinality of the action space. We call this method equivalent effect abstraction. On MDP homomorphism benchmarks, we demonstrate improved sample efficiency over previous attempts to learn MDP homomorphisms, while achieving comparable sample efficiency to approaches that rely on prior knowledge of environment symmetries.

## 1 Introduction

Modern reinforcement learning (RL) agents outperform humans in previously impregnable benchmarks such as Go (Silver et al., 2016) and Starcraft (Vinyals et al., 2019). However, the computational expense of RL hinders its deployment in promising real world applications. In environments with large state spaces, RL agents demand hundreds of millions of samples (or even hundreds of billions) to learn a policy—either within an environment model (Hafner et al., 2020) or by direct interaction (Mnih et al., 2013). Function approximation can enable some generalisation within a large state space but still most RL agents struggle to extrapolate value judgements to equivalent states.

In contrast, animals can easily abstract away details about states that do not effect their values. For example, a foraging mouse understands that approaching a goal ~~state (shown as a piece of cheese in Figure 1)~~ while travelling east will have the same value as approaching the same goal ~~state~~ from the west. These sort of state abstractions have been defined in RL as Markov decision process (MDP) homomorphisms Ravindran & Barto (2001); van der Pol et al. (2020b). MDP homomorphisms reduce large state-action spaces to smaller abstract state-action spaces by collapsing equivalent ~~state-action pairs~~ state-actions in an observed MDP onto a single abstract state-action pair in an abstract MDP van der Pol et al. (2020b).

Given a mapping between an abstract MDP and an experienced MDP, policies can be learned efficiently in the smaller abstract space and then mapped back to the experienced MDP when interacting with the environment (Ravindran & Barto, 2001). However, obtaining mappings to ~~and from~~ the abstract state-action space is challenging. ~~Success has been achieved by hard coding homomorphisms into policy networks~~ Previous works hard code homomorphisms into a policy (van der Pol et al., 2020b) but learning homomorphic mappings online is an unsolved problem.

We develop equivalent effect abstraction, a method that constructs MDP homomorphisms from experience via a dynamics model—leveraging the fact that state-action pairs leading to the same next state usually have equivalent value. Consider the example of a gridworld maze, moving to a given cell has the same value whether you approached from the right or the left. Therefore, if we know the

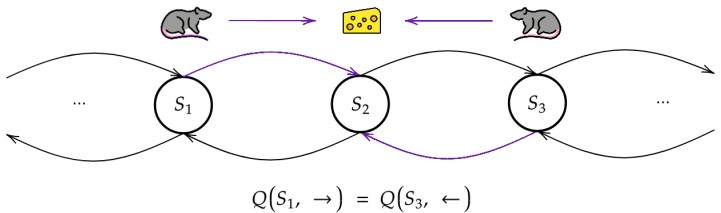

$$Q(S_1, \rightarrow) = Q(S_3, \leftarrow)$$

Figure 1: State-action pairs that lead to the same state often have equivalent values (shown by the purple arrows). Instead of learning these equivalent values individually through reinforcement, we instead learn the value of one abstract action that represents both purple arrows. These values are looked up and learned with a backwards dynamics model during training—meaning no prior knowledge is required from a practitioner.

value for approaching from the right then we also know the value of moving the approaching from the left. By extrapolating value judgements between equivalent state-action pairs we can use this equivalence to reduce the amount of experience required to learn a policy.

An important distinction from previous works is that *we do not use environment symmetries* to reduce the size of the state-action space. We exploit a separate redundancy common to many MDPs—for a given state there are often multiple state-action pairs that also lead to that state. While we do use a partial model in our approach, equivalent effect abstraction is different to model based RL because we focus on reducing the size of the state-action space rather than augmenting experience with predicted trajectories. Additionally, unlike model-based RL, equivalent abstraction can be plugged into model-free algorithms without a reward model. Our contributions are as follows:

1. We develop a novel approach for constructing MDP homomorphisms (equivalent effect abstraction) that requires no prior knowledge from a practitioner

2. In the tabular setting, we show equivalent effect abstraction can improve the planning efficiency of model-based algorithms and the sample efficiency of model-free algorithms

3. In the deep RL setting, we show equivalent effect abstraction can be learned from experience and then leveraged to improve sample efficiency

In Section 2 we formally describe ~~the MDP homomorphism framework~~ MDP homomorphisms and then introduce equivalent effect abstraction in Section 3. In Section 4 we ~~empirically~~ validate our approach using benchmarks from the MDP homomorphism literature. ~~An overview of related work is found in Section 5 and we finish with limitations in Section 6 as well conclusions in Section~~ Related work, limitations and conclusions are in Sections 5, 6 and 7.

## 2 MDP HOMOMORPHISMS

Using the definition from (Silver, 2015), an MDP $\mathcal{M}$ can be described by a tuple $\langle \mathcal{S}, \mathcal{A}, \mathcal{P}, \mathcal{R}, \gamma \rangle$ where $\mathcal{S}$ is the set of all states, $\mathcal{A}$ is the set of all actions, $\mathcal{R} = \mathbb{E}[R_{t+1}|S_t = s, A_t = a]$ is the reward function that determines the scalar reward received at each state, $\mathcal{P} = \mathbb{P}[S_{t+1} = s'|S_t = s, A_t = a]$ is the transition function of the environment describing the probability of moving from state to another for a given action and $\gamma \in [0, 1]$ is the discount factor describing how much an agent should favour immediate rewards over those in future states. An agent interacts with an environment through its policy $\pi(a|s) = \mathbb{P}[A_t = a|S_t = s]$ (Silver, 2015) which maps the current state to a given action. To solve an MDP, an RL agent must develop a policy that maximises the return $G$, which is equal to the sum of discounted future rewards $G = \sum_{t=0}^{T} \gamma^t R_{t+1}$ (where $t$ is the current timestep and $T$ is the number of timesteps in a learning episode). It is worth briefly mentioning that equivalent effect abstraction assumes an MDP definition where the reward function is defined by a given state rather than how an agent travels to that state (i.e. reward functions are defined as $\mathcal{R}(s')$ rather than $\mathcal{R}(s, a, s')$—in the vast majority of RL benchmarks this is a safe assumption.

(Ravindran & Barto, 2001) introduced the concept of a homomorphism which, using the notation and definitions from (van der Pol et al., 2020b), is a homomorphism $h = \langle \sigma, \{\alpha_s | s \in \mathcal{S}\} \rangle$ from an agent's

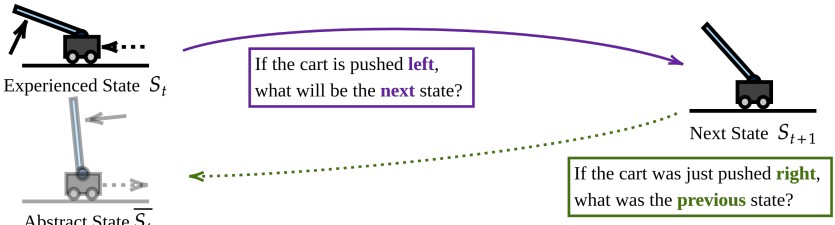

Figure 2: We find equivalent states by moving forward and then backwards through a model of the environment. Unlike previous approaches to homomorphisms in MDPs (van der Pol et al., 2020b), we are *not* exploiting the horizontal symmetry of the environment. In contrast, we take advantage of the fact that state-action pairs that lead to the same state usually have equivalent values.

experienced MDP $\mathcal{M}$ to an abstract MDP $\bar{\mathcal{M}} = \langle \bar{\mathcal{S}}, \bar{\mathcal{A}}, \bar{\mathcal{P}}, \bar{\mathcal{R}}, \gamma \rangle$, which satisfies the following.

$$\bar{\mathcal{P}}(\sigma(s')|\sigma(s), \alpha_s(a)) = \sum_{s'' \in \sigma^{-1}(s')} \mathcal{P}(s''|s, a), \quad \forall s, s' \in \mathcal{S}, a \in \mathcal{A} \tag{1}$$

$$\bar{\mathcal{R}}(\sigma(s), \alpha_s(a)) = \mathcal{R}(s, a), \quad \forall s \in \mathcal{S}, a \in \mathcal{A} \tag{2}$$

Where $\sigma$ is a mapping from experienced states to abstract states, while $\alpha_s$ is a state dependent mapping between experienced and abstract actions. Equations (1) and (2) demonstrate that a homomorphic map of states and actions must maintain the transition dynamics and reward functions of the experienced MDP within the constructed abstract MDP. As proved in (Ravindran & Barto, 2001) and leveraged more recently by (van der Pol et al., 2020b), if the true Q-values in the abstract MDP are known — $\bar{Q}^*(\sigma(s), \alpha_s(a))$ —then equipped with a homomorphism $h$, the true Q-values in the experienced MDP can be "lifted" (Ravindran & Barto, 2001) from their abstract counterparts.

$$\bar{Q}^*(\sigma(s), \alpha_s(a)) = Q^*(s, a) \quad \forall s \in \mathcal{S}, a \in \mathcal{A} \tag{3}$$

Where $Q$ is the value of observing state $s$ and selecting action $a$ and $*$ signifies the value function is optimal (van der Pol et al., 2020b). ~~This is useful because the~~ The size of abstract state-action space $\bar{\mathcal{S}} \times \bar{\mathcal{A}}$ is ~~usually much~~ often smaller than the ~~size of the~~ experienced state-action space $\mathcal{S} \times \mathcal{A}$. Consequently, if an agent is equipped with a homomorphic map, a policy can be learned efficiently in the ~~reduced~~ abstract state-action space and then "lifted" to the experienced state-action space ~~when interacting with the environment (Ravindran & Barto, 2001). Unfortunately, without knowing an environment's symmetries,~~ (Ravindran & Barto, 2001). In general, obtaining a homomorphic map is difficult. ~~In the next section~~ Next, we present a novel method for inferring a homomorphic map from experience.

## 3 Equivalent Effect Abstraction

Equivalent effect abstraction is based on a few simple observations that are guaranteed to be true in most RL tasks if $\mathcal{R}$ and $\mathcal{P}$ are deterministic. Firstly, state-action pairs that lead to the same next state have equivalent reward functions ~~(by definition).~~ in the majority of RL benchmarks (e.g. Deepmind control Tassa et al. (2018), gym control environments Brockman et al. (2016) and the Atari learning environment Bellemare et al. (2013)). While common MDP formulations (like the one used in Section 2) define reward functions as depending on the current state *and* previous action, in many environments it is possible to drop the dependence on the previous action. Secondly, state-action pairs that lead to the same next state also have equivalent transition functions under an unknown state homomorphism function $\sigma$. To leverage these properties to learn policies in an abstract MDP, we must develop a method to learn the mapping $\sigma$ between state-action pairs that lead to the same state.

Consider the normal RL scenario, where an agent takes an action $a$ in a state ~~represented by a vector~~ $s$ and ends up in state $s'$. Once in state $s'$, how can we infer what other state-action pairs could also lead to state $s'$? We could make this inference if we had

---

**Algorithm 1** Equivalent Action Abstraction Q-Network

---

```python
class EquivalentEffectQNetwork(nn.Module):
    def __init__(self, forward_model, backward_model, num_of_actions, canonical_action):
        self.forward_model = forward_model # pretrained forward model
        self.backward_model = backward_model # pretrained backward model
        self.number_of_actions = num_of_actions
        self.canonical_action = canonical_action
        self.q_net = nn.Sequential(nn.Linear(2, 250), nn.ReLU(), nn.Linear(250, 1))

    def forward(self, state):
        values = [] # list to store q-values for each action
        for i, an_action in enumerate(self.number_of_actions):
            next_state = self.forward_model(state, an_action)
            equivalent_state = self.backward_model(state, canonical_action)
            value_for_ith_action = self.q_net(equivalent_state)
            values.append(value_for_ith_action)
        return values
```

---

Figure 3: Pytorch (Paszke et al., 2017) style pseudo-code for equivalent effect abstraction Q-network. Our Q-network only has an output for one action—the canonical action. To transform input states into the canonical reference frame we move forwards and backwards through a dynamics model, with the input action on the forwards model giving the order of the equivalent Q-value outputs.

a forwards *and* backwards model of the environment. To train our forward model $\mathcal{F}$ and backwards model $\mathcal{B}$, we collect experience tuples $\langle \mathbf{s}, a, r', \mathbf{s}' \rangle$ from the environment and then optimise the parameters of our models $\theta$ and $\phi$ with an MSE loss function.

$$\arg\min_{\theta} \|\mathcal{F}_\theta(\mathbf{s}, a) - \mathbf{s}'\|_2^2 \qquad (4) \qquad\qquad \arg\min_{\phi} \|\mathcal{B}_\phi(\mathbf{s}', a) - \mathbf{s}\|_2^2 \qquad (5)$$

With a learned forwards model $\mathcal{F}_\theta$ and backwards model $\mathcal{B}_\phi$ we are ready to reduce the size of our MDP. First, we select an action $\bar{a}$ to be our *canonical* action. Next for a given state-action pair $(\mathbf{s}, a)$ the equivalent (canonical) state-action pair can be computed by moving forwards through our model and then backwards with the canonical action $\bar{a}$.

$$\bar{\mathbf{s}} = \sigma(\mathbf{s}, \bar{a}) = \mathcal{B}_\phi(\mathcal{F}_\theta(\mathbf{s}, a), \bar{a}) \qquad (6)$$

We describe the combination of the forwards and backwards model as $\sigma$ solely to be consistent with the previous literature van der Pol et al. (2020b) (i.e. it is a state homomorphism). In simple terms, $\sigma$ is a mapping to another state action pair that leads to the same next state. To construct $\sigma$, we combine a forwards model prediction and a backwards model prediction. The first model is conditioned on the action $a$ you would like to know the value for and the reverse action $\bar{a}$ is the canonical action.

In theory, one could calculate multiple equivalent state-action pairs for a given state-action pair—by querying the backwards model with multiple different actions. However, our aim is to simplify the state-action space that our policy learns in, so instead we select one action to be our "canonical" action and all other state-action pairs are mapped into its reference frame. In Figure 2, the canonical action is moving right—meaning our Q-network only has to learn about moving right and we can map all right values to their equivalent left values with our forwards and backwards models. The canonical action is a hyperparameter. In some environments such as Cartpole, the canonical action can be selected randomly. However, in other environments backwards state predictions are not always possible for a given state-action pair—meaning canonical actions need to be selected more carefully. Next, we discuss situations where backwards predictions cannot be made.

By finding an equivalent canonical state-action pair for each experienced state-action pair we reduce the number of state-action pairs that an off-policy algorithm must learn to ~~estimate—effectively reducing~~ estimate—reducing the size of the space of values from $\mathcal{S} \times \mathcal{A}$ to ~~$\mathcal{S} \times 1$.~~$\mathcal{S} \times 1 + N$, where $N \in \mathcal{S} \times \mathcal{A}$ are the states where the canonical action is not computable because of the edge cases described in Section 2 (e.g. stochastic environments, no unique backwards action or transitions that do not exist in the environment such as traveling left to a state with a border to its right).

It is important to stress that with forwards and backwards models we have obtained a mapping $\sigma$ that satisfies Equation 3. This mapping $\sigma(\mathbf{s}, \bar{a})$ can map any state-action pair into a smaller state-action space—matching it with a state-action pair that has equivalent value Ravindran & Barto (2001). As our mapping satisfies Equation 3, it also assumes the MDP homomorphism properties developed by Ravindran & Barto (2001), meaning the policy we learn with our canonical state action pairs can be "lifted" van der Pol et al. (2020b) and used as a policy in a given experienced environment.

To leverage this property in Q-learning, for every state-action pair we want to know a value for we ~~simply~~ apply Equation (6) to find the equivalent state and ~~always use the canonical action as our equivalent action (i. e. $\alpha_s = \{\bar{a}, \forall a \in \mathcal{A}\}$). Algorithm ?? describes the full modified Q-learning algorithm.~~ then predict its value with a Q-network. Our definition of an MDP homomorphisms is slightly different from that introduced by (Ravindran & Barto, 2001). Equivalent effect abstraction considers equivalent transitions that collide into the same future state with the same reward. This means it is difficult to map equivalent effect abstraction onto the definitions in Equations (1) and (2). Nevertheless, the most pertinent property of MDP homomorphisms introduced by (Ravindran & Barto, 2001) is that the state-action homomorphism is invariant to value (see Equation (3)), which we demonstrate with our empirical results.

~~Lastly, we~~ We have assumed that for every state-action pair, at least one equivalent canonical state-action ~~exists, which~~ can be computed. Given a model of the environment, this is often true but ~~not guaranteed~~it is not guaranteed for a variety of reasons. For example, near borders in a grid world there is no way to travel left to a border state that has a border on its ~~right—we discuss this in more detail in Sections 4.4 and 6~~right. We only found this to significantly effect results in the a tabular setting, which we mitigate by reverting to vanilla Q-learning table when necessary (that is ~~leaned~~ learned in parallel). Furthermore, stochasticity could be problematic with the current formulation. If a transition is multimodal and the models predict the mean of the two modes then the value network will be presented with states that do not actually exist in the environment. In Section 6, we discuss how recurrent state space models could be used to sample from a distribution of predicted transitions in future work (Hafner et al., 2019). Lastly, if forward dynamics are deterministic the previous state could still be unpredictable as a unique previous state for a given previous action may not exist—again the model based RL provides some ideas for addressing this in future work (Yu et al., 2021).

## 4 EXPERIMENTS

We ~~validate~~ test our approach on a ~~wide~~ range of RL tasks, from tabular ~~reinforcement learning~~ RL with no function approximation to using a convolutional DQN Silver et al. (2016) to do control from pixels. Where possible, we try to have overlap in our experiments with the MDP homomorphism literature—using Cartpole to overlap with the well known van der Pol et al. (2020b) and van der Pol et al. (2020a) as well as Predator Prey to overlap with van der Pol et al. (2020a). Hyperparameter settings and sweep configurations can be found in Appendix A.1. Shaded regions indicate standard error of the mean.

### 4.1 TABULAR MAZE

We begin with the maze environment from (Sutton & Barto, 2018, p. 165). The maze consists of $6 \times 9$ cells. The agent starts on the far West of the maze and must navigate East and find its way around borders in the middle of the environment. After passing the borders, the agent must then travel further East to reach a corridor, before moving North to reach a goal location. In this particular experiment, we assume a model of the environment is known beforehand—which we move through forwards and then backwards to create a homomorphic map.

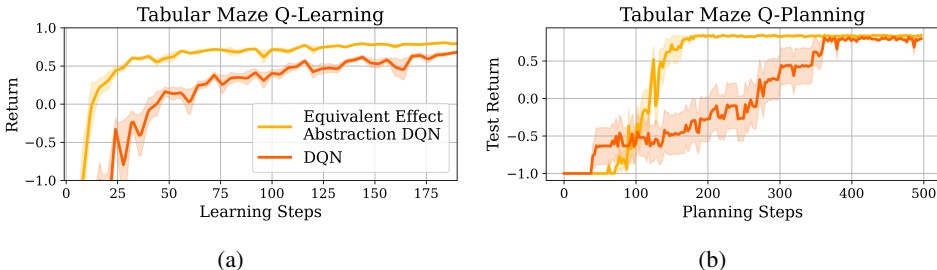

(a) (b)

Figure 4: $(\mathbf{a}), (\mathbf{b})$ In a tabular gridworld abstracting away redundancies in the Q-table improves sample efficiency. For the vanilla Q-learning/planning baselines, an agent must learn the value of each state-action pair for all actions in the action space. With equivalent effect abstraction, we reduce the number of Q-values that need to be learned. This yields improvements in both model-free methods (Q-learning) and model-based approaches (Q-planning). 50 seeds are used for Q-learning while 10 seeds are used for Q-planning.

We use an open-source Q-learning (Watkins & Dayan, 1992) implementation as a baseline[1] (along with the maze code from the same authors). We then adapt the Q-learning implementation to make use of a homomorphic map as shown in Algorithm ??. 1. We show in Figure 4a that our approach compares favorably to vanilla Q-learning, converging much faster to the optimal policy. To demonstrate the potential for improved In the gridworld environment, we reduce the size of the state action space and hence the number of Q-values (when going left) from 216 to 67. In this case, the number of states in the reduced action space is not reduced by the cardinality of the action space exactly. This is because of the 13 states where it is not possible to reach them by traveling left because there is a border to the right.

To test planning efficiency, we performed a further experiment on the gridworld with model-based Q-planning (Sutton & Barto, 2018, p. 161). Our Q-planning method is equivalent to the Q-learning implementation but altered to learn by randomly sampling model transitions. At each Q-planning update we pause training and evaluate performance in a 100 step episode. Planning performance is shown in Figure 4b. Similarly to the Q-learning results, leveraging knowledge of equivalent actions is able to significantly improve learning efficiency.

## 4.2 CARTPOLE

Next, we apply our approach to the Cartpole benchmark from test our approach on Cartpole Brockman et al. (2016). In Cartpole, an agent controls a cart with a pendulum attached to it. The goal of the task is to learn to balance the pendulum upright by moving the cart horizontally left or right. The environment is formulated with inputs represented by the states are a four-dimensional vector (position, velocity, angle, angular velocity) and Carpotle has a discrete action space of moving (move the cart left or right). To learn a model, we use 3 episodes of learning experience at the beginning of training . With these 3 episodes we are able to train the homomorphic map (two simple linear models, one for each action optimised with Adam (Kingma & Ba, 2015)) to generate equivalent state-action pairs for the opposite action to the one experienced. These initial training steps *are* included for equivalent effect abstraction in Figure 5b. After these initial episodes we freeze the learned mapping.

We integrate the learned homomorphism into a DQN implementation Silver (2015) [2], with Q-values only being learned for an arbitrarily chosen canonical action. We compare our approach to three baselines: vanilla DQN, MDP Homomorphic Networks van der Pol et al. (2020b) and PRAE van der Pol et al. (2020a). For this experiment and those in future sections, all experiments we use the baselines' open source implementations and the libraries they build upon Stooke & Abbeel (2019). MDP Homomorphic networks uses specially constructed weights that are equivariant to

---

[1] https://github.com/thehawkgriffith/dyna-maze
[2] https://gist.github.com/Pocuston/13f1a7786648e1e2ff95bfad02a51521

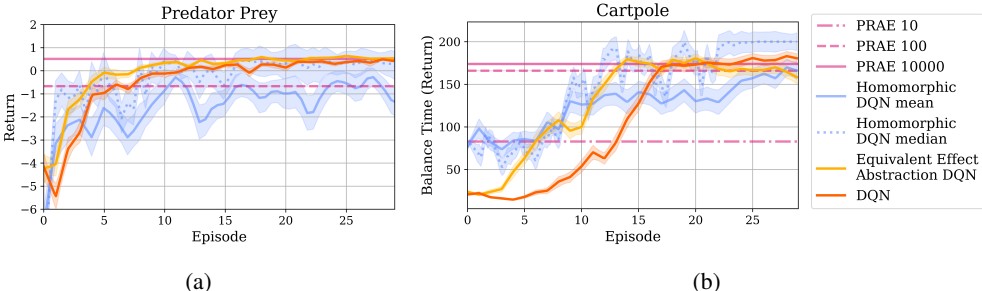

(a)                                                                                    (b)

Figure 5: (**a**) In ~~the stochastic~~ predator prey~~environment~~, the homomorphic DQN is able to learn a mapping to improve sample efficiency over vanilla DQN. ~~Note here that unlike~~ Unlike in Cartpole *the experience used to train the homomorphic map is not included in the plot* for both the equivalent effect abstraction DQN. In the case of equivalent effect abstraction we reuse one set of pre-trained backwards and forwards models for each RL training run. (**b**) In Cartpole, the improvement in sample efficiency is more dramatic~~, with equivalent effect abstraction significantly improving convergence speed~~. Equivalent effect abstraction learns a model for improved convergence very quickly (the model is learned in the initial 3 episodes of experience *that is shown on the x-axis of the plot*). We retrain this model for every repeat. For vanilla DQN and equivalent effect abstraction we use 50 seeds, 5 seeds are used for each PRAE result and 30 seeds are used for MDP homomorphic networks. For clarity, PRAE 10 refers to using PRAE with 10 episodes of data to construct an environment model, which it then plans in.

environment symmetries ~~to improve sample efficiency~~ (requiring prior knowledge of symmetries). PRAE trains a contrastive model to learn a mapping to a latent "plannable" MDP, that satisfies the definitions of an MDP homomorphism. PRAE then performs planning on the learned abstract MDP. For Cartpole, we tried to learn a mapping for PRAE with datasets of 10, 100, 10000 episodes of random experience. We plot the average convergence performance of PRAE's planning algorithm. In the cartpole task, we did not need to avoid any canonical state action pairs that are not computable—meaning a reduction of the size of the state action space from $\mathcal{S} \times \mathcal{A}$ to $\mathcal{S} \times 1$.

As shown in Figure 5b, in the low sample regime we improve upon both PRAE and vanilla DQN—with our approach converging at around episode 12 while vanilla Q-learning takes around 16 episodes to converge—note that this improvement includes the number of episodes required to learn our mapping. We found that in many cases MDP homomorphic networks were able to achieve good performance but worst case runs brought down mean performance significantly. To demonstrate this, we plot the mean and median for MDP homomorphic networks (only the mean is plotted for other methods). The median MDP homomorphic network is also able to similarly improve upon the vanilla DQN baseline, by leveraging a practitioner's prior knowledge of environment symmetries. MDP homomorphic networks are not necessarily a competing approach to equivalent effect abstraction and future work could conceivably combine these two approaches for even greater sample efficiency.

### 4.3 STOCHASTIC PREDATOR PREY

Following the seminal work of van der Pol et al. (2020b), we also benchmark our method on the predator prey environment, where an agent ~~much~~ must chase a stochastically moving prey in a 2D world van der Pol et al. (2020a). The observations are a $7 \times 7 \times 3$ tensor encoding agent position and prey position. The objective of the agent is to catch the prey in fewest steps possible. Rewards are set to -0.1 unless the predator catches the prey, in which case the episode ends and a reward of 1 is provided.

For equivalent effect abstraction, we train an action dependent forwards and backwards models on an experience dataset created by taking random actions in the environment for $10^4$ environment steps (equivalent to around 170 episodes of random experience). An interesting point to note here is that the environment is stochastic, so learning a perfect model of the environment is impossible. We compare to the same baselines introduced in Section 4.2. For PRAE, we benchmark with both 10000 and 100 episodes of experience data and then perform planning to convergence. ~~Equivalent effect abstraction delivers~~ If a model can be obtained before online

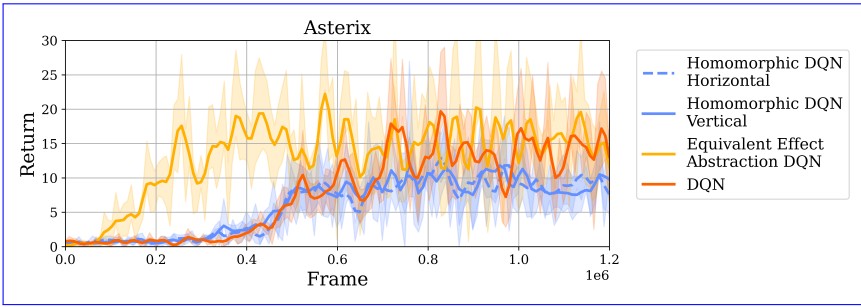

Figure 6: In the more challenging environment of Asterix, Equivalent effect abstraction allows DQN to reach a return of 20 in around $(3.5 \times 10^5)$ frames, while vanilla DQN requires approximately $(1 \times 10^6)$ frames to reach this return. We run each method for five seeds. For readability, both plots have been smoothed with a Gaussian kernel where $\sigma = 1$.

policy learning, then equivalent effect abstraction can deliver an improvement over vanilla DQN ~~, consistent with the Cartpole results.~~ on the stochastic predator prey environment. However, for this environment, the number of episodes to learn a policy is extremely small ($\sim 5$), so learning a model of the environment online is not practical. As a policy can be learned very quickly, it is not a particularly useful environment to test sample efficiency—it instead serves as a useful benchmark (as demonstrated in van der Pol et al. (2020b)) to address the possible limitations of MDP homomorphisms in stochastic environments. Again in a similar vein to Cartpole, PRAE can be very effective, but only when a relatively large amount of model training experience is available.

## 4.4 MINATAR ASTERIX

MinAtar Young & Tian (2019) emulates a subset of games from the Atari learning environment Bellemare et al. (2013) but at a lower dimensionality—allowing policies similar to those required in the full Atari learning environment to be trained at lower computational ~~cost. This is useful because we~~ cost—we can do similar benchmarking to previous works on MDP homomorphisms (i.e. van der Pol et al. (2020b)'s Atari experiments) but within tasks that are easy to reproduce. We use *Asterix* to benchmark equivalent effect abstraction. In Asterix, the agent must explore its environment and capture moving treasure pots, whilst avoiding being struck by moving enemies~~. Visualisations of the environments can be found here~~[3]. For Asterix, the effective actions are left, right, up, down and "do nothing". It is important to note that Asterix has no obvious global symmetries, making it difficult for approaches that rely on symmetries van der Pol et al. (2020b).

In this section, we assume a model of the environment is known, which we construct programmatically to perform equivalent effect abstraction (see supplementary code). Thus the question addressed here is *not* about learning backwards and forwards models as we did for Cartpole (section 4.2) and Predator Prey (section 4.3). Instead we seek to understand, given a model of the environment, what approach can we take to perform value based deep RL in the smallest number of learning steps.

We use the convolutional DQN Silver et al. (2016) implementation from the Minatar repository Young & Tian (2019) as the backbone for equivalent effect abstraction, leaving their hyperparameters unchanged. As is the case in many games (and RL environments more generally), MinAtar allows a "do nothing" action. Interestingly, it is always possible to construct our homomorphism with "do nothing" as our canonical action, meaning we never have to fall back to vanilla Q-learning (which we did previously in the tabular maze when the homomorphism was impossible to construct). In Asterix it is always possible to reach every state by doing the canonical action—meaning we reduce the size of the state-action space by a factor of 6 (the number of actions in the action space). We compare equivalent effect abstraction to the vanilla DQN from the MinAtar Young & Tian (2019) repository (again leaving hyperparameters unchanged). As a further baseline, we construct a DQN with a equivariant feature extractor, similar to the network used to learn Pong in van der Pol et al. (2020b). We implement feature extractors that are equivariant to both horizontal

---

[3]https://github.com/kenjyoung/MinAtar/blob/master/README.md

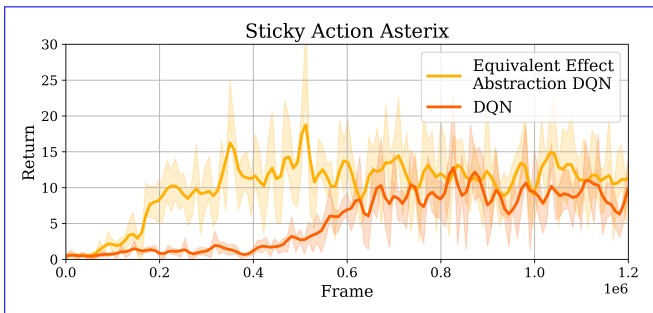

Figure 7: Equivalent effect abstraction maintains its advantage over a vanilla DQN network in the presence of stochastic transitions. Following (Machado et al., 2018) we perform additional experiments using the "sticky actions" protocol. We supply our agent with a perfect model in the non-stochastic setting, meaning 25% of the time its predictions are incorrect (due to stochasticity). We also perform linear interpolation to sync frame and return points due to logging quirk with wandb (Biewald, 2020), see Appendix B for details.

and vertical reflections. The hyperparameter selection process for this baseline is described in A.1.4.

Asterix is a relatively complex game requiring an agent to explore for moving treasures while simultaneously avoiding enemies~~(who are also moving)~~. During this exploration~~for treasure~~, we show empirically that traversing the state space is much more efficient when the equivalences between different state-action pairs are known. Equivalent effect abstraction converges $3\times$ faster than vanilla Q-learning, consistent with out results in previous sections. As there is no obvious symmetry—as there is for Pong for example—homomorphic DQN does not yield an advantage over the vanilla DQN.

### 4.4.1 STICKY ACTIONS

The original Minatar Asterix has some elements of stochasticity for one step predictions. For example, with one step predictions one cannot predict when spirtes will appear at the side of the screen. Nevertheless, once the sprites appear, their movements are deterministic. This is an issue also present in the full high-dimensional atari emulator Machado et al. (2018), where an agent can overfit to fully deterministic transitions. As a result, Machado et al. (2018) developed the sticky actions protocol where 75% of the actions given by the agent's policy are delivered to the emulator, while 25% of the time the emulator uses the agents previous action. We apply this approach to our Asterix experiments to test how well equivalent effect abstraction can cope with stochastic environments. We use the same environment model as we did in the deterministic environment and show that equivalent effect abstraction can maintain its advantage over the vanilla DQN even when transitions are unpredictable.

## 5 RELATED WORK

(Ravindran & Barto, 2001) developed a ~~theoretical framework for state-action abstraction under symmetry, while also deriving~~ framework for abstraction under symmetry—deriving theoretical guarantees when lifting abstract policies to the experienced MDPs. (Givan et al., 2003) proposed MDP model minimisation using bisimulation (Larsen & Skou, 1991), where states could be mapped to ~~a common~~ an abstract state if their expected transition dynamics and reward sequences were indistinguishable. ~~Despite their nice theoretical properties,~~ These early applications of MDP homomorphisms ~~were limited because they required~~ required of prior knowledge from a practitioner~~and there was also no obvious method for integrating homomorphisms into function approximation. (Biza et al., 2021) also~~. (Biza et al., 2021) use bisimulation to inform structured Hidden Markov Model priors to infer an explicit discrete reduced state space from learnt embeddings in deep RL. They show ~~that policies resulting from planning in~~ polices from the reduced abstract space are viable in the original state space.

It has also been shown that bisimulation can be relaxed from a binary mapping to a similarity metric in the works of (Ferns et al., 2011; Ferns & Precup, 2014)—these ~~bisimulation~~ metrics have been used to condition embeddings using contrastive losses to generate invariant representations of states~~in deep RL~~. (Zhang et al., 2020) augment the embedding encoder of their model with an additional contrastive loss term that, when used in conjunction with RL, results in an encoder in which the distance between processed observations in embedding space corresponds to an approximate bisimulation ~~metric. This allows~~ metric—allowing agents to perform control tasks in the presence of distractor cues~~being inserted into their observations~~. A more general similarity metric for conditioning embedding in deep RL was proposed in Agarwal et al. (2021), where they outlined a new policy similarity metric. This metric is also used in conjunction with a contrastive loss, and pulls together embeddings that result in similar policies and long term behaviours. ~~While using contrastive learning and bisimulations has achieved impressive results , contrastive methods~~ Contrastice learning achieves impressive results but can be sample hungry—making their deployment in the low sample regime difficult van der Pol et al. (2020a)).

More recently, homomorphisms in MDPs have been promoted as a means to improve sample efficiency in tasks with symmetries. PRAE (van der Pol et al., 2020a) trained a contrastive world model and accompanying loss function that satisfies the transition dynamics definition (Equation (1)) by design—with good results in Cartpole when 1000 episode of experience are available for training. However, in the relatively low sample regime (e.g. 100 episodes) their homomorphism is less effective as we show in Figure 5b. (van der Pol et al., 2020b) derived a method for learning network weights that are equivariant to environment symmetries (which implicitly creates an MDP homomorphism) but their algorithm requires a practitioner to hardcode symmetric groups beforehand. Similar approaches have been adopted for environments with continuous symmetries Wang et al. (2022). Another recent approach to using MDP homomorphisms is detailed in (Biza & Platt, 2019). Here the problem of finding MDP homomorphisms is approached using Online Partition Iteration—the mappings from state to abstract states are learnt by predicting which partition a state should fall into given an action, and refining the partitions through splitting.

Perhaps the most relevant related approach is the use of afterstates Sutton & Barto (2018)[p. 136]. Afterstates slightly shift an MDP out of phase with conventional state transitions—creating environment states that are in-between the initial effect of a policy's action and the reaction of the environment to said action. However, applications of the afterstates framework have generally been constrained to board games Tesauro et al. (1995) and usually focus on dealing with the stochasticity of an opponent rather than improved sample efficiency Antonoglou et al. (2021). (Misra et al., 2020) use contrastive learning to learn a similar abstraction—grouping together state-actions that will pass through or have passed through same state.

More broadly, model-based RL has enabled superhuman performance in Atari with a ~~relatively~~ small amount of experience (Hafner et al., 2020), but the number of planning steps required to learn a policy is still very large. Backwards models have ~~recently~~ been proposed to improve sample efficiency of world model representation learning, which would be interesting if integrated with equivalent effect abstraction (Yu et al., 2021). A related exploration method shows simple autoencoders can learn binary abstract representations useful for logging of what states an agent has experienced (Tang et al., 2017). Our work is related to approaches that learn data augmentation policies for RL (Raileanu et al., 2021), however instead of ~~trying to augment~~ augmenting experience with a more diverse distribution of states, we ~~take a different approach of~~ are trying to learn values for a narrower distribution of canonical states.

## 6  LIMITATIONS AND FUTURE WORK

Equivalent effect abstraction requires a ~~competent model of environment dynamics~~ dynamics model. For equivalent effect abstraction to be practical, the number of ~~environment steps~~ transitions required to learn the environment model must be smaller than ~~the number steps needed~~ those required to learn a ~~policy—luckily this~~ policy—this is true in many environments (e.g. Cartpole) but it ~~cannot be~~ is not guaranteed. What's more, despite recent progress ~~in model based learning in high dimensional environments~~ Hafner et al. (2019; 2020); Saxena et al. (2021); Ye et al. (2021), learning a dynamics model can be difficult to learn for a number of reasons. Firstly, stochastic dynamics can make

environment transitions multimodal, meaning point predictions of future or previous states are not adequate. In theory, this can be dealt with using stochastic models Hafner et al. (2019) of forwards and backwards dynamics that could be sampled when abstracting state-action pairs. Even worse, dynamics can be completely unpredictable Kendall & Gal (2017); Mavor-Parker et al. (2021), in which case ~~aleatoric~~ uncertainty predictions could be used to signal that equivalent effect abstraction should temporarily revert to vanilla Q-learning.

Furthermore, backwards transitions ~~could also be impossible to predict~~ may be unpredictable because an equivalent state does not ~~exist—as described in~~ exist—see Section 1. In this case, uncertainty predictions could also be used to avoid using mappings that are out of distribution ~~with the model training data~~ (because they are not possible) Kendall & Gal (2017). Integrating the described probabilistic transition models into equivalent effect abstraction is an interesting direction for future work. Lastly, ~~although the challenges surrounding model learning are significant,~~ it is worth mentioning that we only require one step transition models, avoiding the significant harder challenge of learning multi-step prediction models Saxena et al. (2021).

~~So far, we have only formulated equivalent effect abstraction~~ Equivalent effect abstraction is formulated within the framework of value based RL ~~in discrete action spaces. While value based methods are behind~~ using discrete actions. While many of the recent breakthroughs in ~~deep RL~~ RL are value based (e.g. Silver et al. (2017), Badia et al. (2020)), actor-critic approaches Schulman et al. (2017); Mnih et al. (2016) are often the natural choice for control tasks. Embedding equivalent effect abstraction into actor-critic architectures is a ~~potentially~~ fruitful avenue for future research. It is also conceivable to formulate equivalent effect abstraction within a continuous action space by simply discretising the action space ~~, which has allowed discrete action space methods to obtain state of the art performance on RL tasks Banino et al. (2021).~~ Banino et al. (2021). Additionally, equivalent effect abstraction could also be integrated into existing homomorphic MDP methods that rely on symmetries to further reduce the size of the abstract state-action space van der Pol et al. (2020b). In practice, this would mean using homomorphic Q-network van der Pol et al. (2020b) but modified to only have one action output as shown in Algorithm 1. Alternatively, planning could be done with an abstract homomorphic representation van der Pol et al. (2020a), which is then reduced further with equivalent effect abstraction.

## 7 CONCLUSION

Equivalent effect abstraction is a simple method that ~~reduces~~ decreases the size of ~~the~~ a state-action space~~considerably for discrete action space MDPs. The approach~~. It is easy to implement and provided with a backwards dynamics model of the environment it requires no prior knowledge of environment symmetries from a practitioner. We have demonstrated that equivalent effect abstraction improves sample efficiency in tabular environments, control tasks with continuous ~~action~~ state spaces, stochastic deep RL environments and also within an image based game playing task. An exciting next step should integrate equivalent effect into popular deep model-based algorithms (Hafner et al., 2020; Yu et al., 2021) to improve planning efficiency. ~~Additionally, equivalent effect abstraction could also be integrated into existing homomorphic MDP methods that rely on symmetries to further reduce the size of the abstract state-action space van der Pol et al. (2020b).~~

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

# A  APPENDIX

## A.1  HYPERPARAMETER SEARCH

Below we show the hyperparameters swept through for homomorphic MDP, DQN and Equivalent Effect Abstraction agents, broken down by environment. The PRAE architectures were generally kept the same as the hyperparameters provided in (van der Pol et al., 2020a), with the exception of the learning rate which we evaluate at $0.0001, 0.001$ and $0.1$

### A.1.1  SUTTON AND BARTO TABULAR GRIDWORLD

We use the hyperparameters specified in (Sutton & Barto, 2018, p. 165): namely, Learning Rate$= 0.1$, $\gamma = 0.95$ and $\epsilon = 0.1$

### A.1.2  CARTPOLE

Table 1: Hyperparameters swept through for the Cartpole environment. Learning rate decay refers to decaying the learning rate by a factor of ten at after a specified number of episodes have elapsed.

| Hyperparameter | Values |
| --- | --- |
| Learning Rate | $0.00001, 0.0001, 0.001, 0.01$ |
| $\epsilon$ decay schedule | No decay, exponential $\tau = \frac{-1}{200}$ |
| $\gamma$ | 0.8, 0.99 |
| Activation | ReLU, tanh |
| Learning Rate decay | No decay, 5, 10, 15, 20 |

**Homomorphic MDP** best hyperparameters: Learning Rate = 0.001, $\epsilon$ decay schedule = No decay, $\gamma = 0.8$, activation = tanh, Learning Rate decay = No decay

**Equivalent Effect Abstraction** best hyperparameters: Learning Rate = 0.001, $\epsilon$ decay schedule = No decay, $\gamma = 0.8$, activation = tanh, Learning Rate decay = 10

**Vanilla DQN** best hyperparameters: Learning Rate = 0.001, $\epsilon$ decay schedule = No decay, $\gamma = 0.8$, activation = tanh, Learning Rate decay = 15

### A.1.3  STOCHASTIC PREDATOR PREY

Table 2: Hyperparameters swept through for the Predator Prey environment.

| Hyperparameter | Values |
| --- | --- |
| Learning Rate | 0.0001, 0.001, 0.01 |
| $\gamma$ | 0.8, 0.99 |

**Homomorphic MDP** best hyperparameters: Learning Rate = 0.001, $\gamma = 0.99$

**Equivalent Effect Abstraction** best hyperparameters: Learning Rate = 0.01, $\gamma = 0.8$

**Vanilla DQN** best hyperparameters: Learning Rate = 0.001, $\gamma = 0.99$

### A.1.4 MINATAR

All hyperparameters were kept at the tuned DQN values provided in Young & Tian (2019). For the Homomorphic MDP network, we use a symmetric group for reflection about the vertical axis when training on Breakout; on Asterix we show results for both reflection about the vertical and horizontal axes.

## A.2 MODEL ARCHITECTURES

### A.2.1 CARTPOLE

---
**Listing 1** Homomorphic MDP Network (van der Pol et al., 2020b)

```
BasisLinear*(repr_in=4, channels_in=1, repr_out=2, channels_out=64)
ReLU() / tanh()
BasisLinear(repr_in=2, channels_in=64, repr_out=2, channels_out=64)
ReLU() / tanh()
BasisLinear(repr_in=2, channels_in=64, repr_out=2, channels_out=1)
```

*`BasisLinear` refers to the symmeterised layers used in (van der Pol et al., 2020b) to create homomorphic networks. This network is identical to the Cartpole network presented in that paper, but with only one output head that outputs state-action values.

---
**Listing 2** Value Network architecture for DQN and Equivalent Effect Abstraction

```
Linear(input_size=4, output_size=1024)
tanh()
Linear(input_size=1024, output_size=1024)
tanh()
Linear(input_size=8, output_size=1024)
tanh()
Linear(input_size=1024, output_size=2)
```

---
**Listing 3** Transition Model Architecture for Equivalent Effect Abstraction

```
Linear(input_size=2, output_size=2)
```

---
**Listing 4** PRAE Architectures (van der Pol et al., 2020a)

```
# state encoder
Linear(input_size=4 ,output_size=64)
ReLU()
Linear(input_size=64, output_size=32)
ReLU()
Linear(input_size=32, output_size=50)
#action encoder
Linear(input_size=54 ,output_size=100)
ReLU()
Linear(input_size=100, output_size=2)
# reward prediction network
Linear(input_size=50 ,output_size=64)
ReLU()
Linear(input_size=64, output_size=1)
```

### A.2.2  PREDATOR PREY

---

**Listing 5** Homomorphic MDP Network (van der Pol et al., 2020b)

```
BasisConv2d(repr_in=1, channels_in=1, repr_out=4, channels_out=4,
filter_size=(7,7), stride=2, padding=0)
ReLU()
BasisConv2d(repr_in=4, channels_in=4, repr_out=4, channels_out=8,
filter_size=(5,5), stride=1, padding=0)
ReLU()
GlobalMaxPool()
BasisLinear(repr_in=4, channels_in=8, repr_out=4, channels_out=128)
ReLU()
BasisLinear(repr_in=4, channels_in=8, repr_out=4, channels_out=128)
ReLU()
BasisLinear(repr_in=4, channels_in=128, repr_out=5, channels_out=1)
```

---

This is again the same network used in van der Pol et al. (2020b), albeit with a different output head.

---

**Listing 6** Value Network Architecture for DQN and Equivalent Effect Abstraction

```
Linear(input_size=441, output_size=1024)
ReLU()
Linear(input_size=1024, output_size=8)
ReLU()
Linear(input_size=8, output_size=1024)
ReLU()
Linear(input_size=1024, output_size=5)
```

---

**Listing 7** Transition Model Architecture for Equivalent Effect Abstraction

```
Linear(input_size=882, output_size=512)
ReLU()
Linear(input_size=512, output_size=8)
ReLU()
Linear(input_size=8, output_size=512)
ReLU()
Linear(input_size=512, output_size=441)
```

---

**Listing 8** PRAE Architectures (van der Pol et al., 2020a)

```
# state encoder
Linear(input_size=441, output_size=64)
ReLU()
Linear(input_size=64, output_size=32)
ReLU()
Linear(input_size=32, output_size=50)
#action encoder
Linear(input_size=54, output_size=100)
ReLU()
Linear(input_size=100, output_size=2)
# reward prediction network
Linear(input_size=50, output_size=64)
ReLU()
Linear(input_size=64, output_size=1)
```

---

### A.2.3  MINATAR

---

**Listing 9** Homomorphic MDP Network (van der Pol et al., 2020b)

```
BasisConv2d(repr_in=1, channels_in=4, repr_out=2, channels_out=16,
filter_size=(3,3), stride=1, padding=0)
ReLU()
```

```
Linear(input_size=1408, output_size=256)
ReLU()
Linear(input_size=256, output_size=6)
```

**Listing 10** Value Network Architecture for DQN and Equivilent Effect Abstraction

```
Conv2d(channels_in=4, channels_out=16, filter_size=(3,3), stride=1, padding=0)
ReLU()
Linear(input_size=1024, output_size=128)
ReLU()
Linear(input_size=128, output_size=6)
```

## A.3 Additional Minatar Results

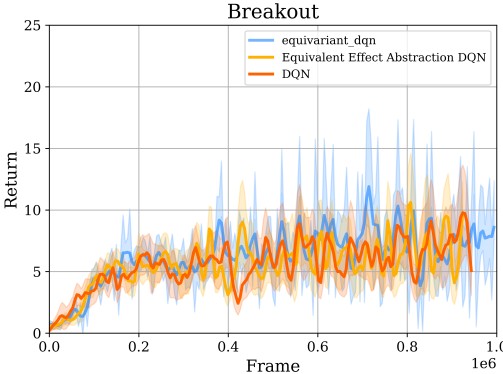

Figure 8: Here we get similar results to van der Pol et al. (2020b) showing that homomorphisms do not necessarily improve sample efficiency in Breakout. On this particular environment all versions of DQN converge almost instantly to a suboptimal policy. To obtain more meaningful results, future work should integrate equivalent effect abstraction into actor-critic methods that are able to solve Breakout Young & Tian (2019).

## B Syncing Log arrays

For the sticky MinAtar results we logged frames and return values with wandb while performing our experiments Biewald (2020). We performed this logging suboptimally as we logged steps for frames and returns separately—meaning we ended up with two arrays where one was NaN when the other was logging a float and vice versa. We used Panda's (pandas development team, 2020) linear interpolator to estimate the NaN values in our final plots. This process does not effect our results significantly as the points were logged at a very high frequency.

