# OpenReview forum: "A Simple Approach for State-Action Abstraction using a Learned MDP Homomorphism"
_ICLR.cc/2023/Conference — Submitted to ICLR 2023_

### Official Review · Reviewer_zsTX · 2022-10-17

**Confidence:** 3
**Correctness:** 2
**Technical Novelty And Significance:** 4
**Empirical Novelty And Significance:** Not applicable
**Recommendation:** 6

**Clarity, Quality, Novelty And Reproducibility:**

I believe the idea is novel and interesting, and certainly appealing due to its simplicity. I have significant doubts about how general it is, though. The empirical verification seems to be reproducible, as all Python code for the models and algorithms has been provided.

**Strength And Weaknesses:**

The paper addresses the problem of automatic abstraction of MDPs, which is a form of representation learning, and is thus very appropriate for the conference. The general approach of looking for equivalence classes is certainly sound, with a long tradition in AI and OR, and the empirical verification looks encouraging. However, I am not sure the chosen criterion for state abstraction is very widely applicable. In Section 3, on page 3, the authors state that "state-action pairs that lead to the same next state have equivalent reward functions (by definition)". This statement is questionable, and would be true only if the reward depends only on the successor state, as in the example with the mouse given. But, this is not how the reward function is defined in Section 2 - there, it depends on both the starting state and the action applied. One other special circumstance where this statement would be true is when the instantaneous reward (respectively, cost) of every transition is the same. This is true for minimum-time optimal control/decision problems, but definitely not always true. Outside of these special cases, the value function of several state-action pairs with the same successor state is not the same, so how could this approach work in such cases?

Furthermore, the method makes use of forward and backward models, which are trained by minimizing prediction errors, according to Equations 4 and 5. This also looks highly problematic. What happens if the transition probability of the original MDP is multimodal for a given state-action pair? It is well known that by minimizing MSE, the model could learn to predict a successor state that is even impossible according to the original transition distribution, because it lies somewhere between the modes of that distribution. If my understanding is correct, all environment the proposed algorithm has been tested on are deterministic, so this would not be a problem for them, but what happens when they are stochastic, with multimodal transition distributions?

The same argument applies to the backward model - in case of stochastic environments, it would be possible to have multiple origin states s that, if action a is applied, will result in the same successor state s'. If all such experienced transitions are used for training the backward model for the pair s',a using MSE, it will learn to predict the average of the origin states s, which might not coincide with any of them. Wouldn't that distort significantly the transition function of the abstracted MDP?



**Summary Of The Paper:**

The paper proposes an approach to MDP state abstraction based on the idea of forming equivalence classes of state-action pairs that must have the same value function, and thus can be limped into a single abstract state, thus improving the sample complexity and speed of solving the resulting abstract MDP. The criterion for grouping state-action pairs into an equivalence class is whether they lead to the same successor state. Empirical verification on several well-known benchmark problems demonstrate that indeed learning on the thus abstracted MDP is faster than on the original one.

**Summary Of The Review:**

I find the idea of using the principle of equivalent effect abstraction quite appealing, but am not convinced that it is applicable to general MDPs with stochastic transition functions. It would be helpful if this is investigated empirically. Nevertheless, even if the method is applicable only to problems with deterministic dynamics, it would still be a valuable addition to the toolbox of practitioners. (But, its limitations should be clearly stated.)

---

> ### Author Response · Authors · 2022-11-16
> **Response to zsTX**
>
> We thank the reviewer for their kind and detailed comments. We are especially excited that zsTX found our work "quite appealing" and that it has the potential to be a "valuable addition to the toolbox of practitioners". We respond to specific concerns below.
>
> **Regarding performance in stochastic environments**
>
> We agree that it is important to understand how our algorithm will perform in stochastic environments. In response to your comments we have performed experiments on Asterix with added stochasticity, following the well-established sticky action protocol for testing how an algorithm performs in non-deterministic environments (Machado et al. 2018). Note that sticky actions make the action selected by the agent be executed 75% of the time, but stochastically the previous action will be executed again instead (see Machado et al. 2018 for specifics)---this makes both the forward and backward predictions stochastic. These results will be added to the updated manuscript.
>
> *We would also like to highlight that the predator prey environment is stochastic as the prey that the predator is chasing randomly moves*. We have updated the manuscript to make this more clear.
>
> **Updating statements about reward functions**
>
> We appreciate your comments demonstrating we were not precise enough in how we define when/how reward functions are defined in terms of states only or on states and previous action. We have made the following updates to our definition to improve clarity:
>
> “Equivalent effect abstraction is based on a few simple observations that are guaranteed to be true in most RL tasks if $\mathcal{R}$ and $\mathcal{P}$ are deterministic. Firstly, state-action pairs that lead to the same next state have equivalent reward functions in the majority of RL benchmarks (e.g. Deepmind control \cite{tassa2018deepmind}, gym control environments \cite{brockman2016openai} and the Atari learning environment \cite{bellemare2013arcade}). While common MDP formulations (like the one used in Section 2) define reward functions as depending on the current state \textit{and} previous action, in many environments it is possible to drop the dependence on the previous action.”
>
> **Action dependent reward functions generally**
>
> Thanks for raising concerns about action dependent reward functions. We recognise that this is a limitation of our algorithm. However, we do not think that this limitation is fatal. For example, common reinforcement learning benchmarks we are aware of (gym classic control tasks, atari, mujoco) only define reward functions in terms of states. Would it be possible to give some examples of RL benchmarks that have reward functions that depend on the current state and previous action?
>
> **Regarding point predictions and Multimodal Transitions**
>
> It is true that our network only performs point predictions of future and previous states. In future work we would like to embed our method within probabilistic models (e.g. Hafner et al. 2020) and sample transition distributions to get equivalent states. We have updated the manuscript significantly to discuss this possibility for future work based on your suggestions. In particular, we discuss multimodal transition distributions below and make your concerns clear to the reader.
>
> “Furthermore, stochasticity could lead to issues with the current formulation of our approach. If a transition is multimodal and the models predict the mean of the two modes then the value network will be presented with states that do not actually exist in the environment \citep{hafner2019learning}. In Section \ref{limitations}, we discuss how recent developments in recurrent state space models could sample from a distribution of predicted transitions in future work \citep{hafner2019learning}. Lastly, even if forward dynamics are deterministic the previous state could be unpredictable as there may not be a unique previous state for a given previous action---again the model based reinforcement learning provides some ideas for addressing this in future work \citep{yu2021learning}.”
>
> In practice for the stochastic environments we have considered (e.g. Predator prey and sticky-action Asterix) we have found that equivalent effect abstraction copes relatively well. For predator-prey we predict the mean next state with our transition models. For Asterix we leave the constructed transition models unchanged but the action input is noisy (i.e. sometimes the environment randomly selects a different action). Equivalent effect abstraction still maintains its improved sample efficiency in both cases (see the updated manuscript in the coming days).

---

> > ### Comment · Reviewer_zsTX · 2022-11-17
> > **Response to authors' response**
> >
> > I appreciate the effort that the authors have put into addressing my comments and clarifying the applicability of their algorithm. Regarding whether the ability to handle action-dependent reward functions is fatal or not, I do agree with the authors that many popular RL problems, including many in the OpenAI Gym, are of this kind. However, it is also true that many classical optimal control problems that are generally amenable to RL algorithms do include a strong dependence of the cost/reward on the action chosen. The simplest example is perhaps the linear quadratic regulator (LQR), where the cost is a quadratic function of the applied effort, and there are many more such problems, typically for applications where some fuel or other type of energy is expended to control the system, and one of the objectives is to minimize it. I think there is nothing wrong with the proposed algorithm, if it cannot handle such cases, as long as this limitation is clearly pointed out.

---

> > > ### Author Response · Authors · 2022-11-25
> > > **response to zsTX**
> > >
> > > Thank you for response and clarifying your concerns. We appreciate you providing an example of the sorts of environments where our algorithm would not be suitable. Based on your suggestions, we have now clearly stated the limitations in Section 3 continued from the previous text:
> > >
> > >
> > > "...While common MDP formulations (like the one used in Section 2) define reward functions as depending on the current state \textit{and} previous action, in many environments it is possible to drop the dependence on the previous action. **Nevertheless, it is important to note that we assume no dependence on the previous action for the reward function. This could be problematic in optimal control settings where different actions have a different cost (e.g. the linear quadratic regulator \citep[p. 145]{liberzon2011calculus}), in which case the current formulation of equivalent effect abstraction would not be suitable.**"

---

### Official Review · Reviewer_MMzC · 2022-10-25

**Confidence:** 4
**Correctness:** 2
**Technical Novelty And Significance:** 3
**Empirical Novelty And Significance:** 2
**Recommendation:** 3

**Clarity, Quality, Novelty And Reproducibility:**

The paper is generally well written and easy to follow, but the limitations of the method are not clearly discussed.

**Strength And Weaknesses:**

Strength
- The idea of aggregating state-action pairs to an equivalent state using a canonical action is very interesting. For some simple problems this can indeed efficiently reduce the size of the equivalent state space for RL algorithms.

- Experiments show that the proposed method could potentially learn faster.

Weakness
- Although the idea of using the forward and backward models makes sense in simple deterministic environments, one cannot get these models for typical stochastic environments. Furthermore, even in deterministic environments, since multiple state-action pairs could lead to the same state, the inverse image of the next state is not a singleton in general. When the inverse image of the system dynamics is not a singleton, it is impossible to learn such a backward model as described in the paper. There may be conditions under which the described forward and backward models exist, but there are no discussions about in which cases is it possible to learn the forward and backward models in the paper.

- The main feature of the approach is to learn a MDP homomorphism with a smaller state space, but there is no analysis on whether the resulting learned model actually provides a simpler MDP homomorphism by either analytical or numerical evaluation. For example, in the tabular maze experiment, what is the equivalent size of state space after the homomorphism map compared with original size?

- Half of the numerical experiments are done with the assumption of the given environment model. It is perfectly fine to consider the situation when the model is given and try to learn a good performing policy. But in this case, comparison with prior methods with the knowledge of model is then necessary, and none is provided in the paper.

- For the predator prey environment, since the proposed method uses 170 episodes to learn the forward and backward models, it is actually not fair to claim that the method outperforms vanilla DQN. When compared with prior homomorphism methods like PRAE, since the number of samples used in learning the homomorphism are different for the methods, one cannot draw the conclusion that the proposed method is indeed more sample efficient.

**Summary Of The Paper:**

The paper proposes a new way to construct homomorphism for MDPs by a forward model and a backward model. Using the forward and backward models and a canonical action, a state-action pair is mapped to an equivalent canonical state-action pair and the effective size of state-action space is reduced. Empirical experiments demonstrate that the proposed method could learn faster than the baselines.

**Summary Of The Review:**

The idea of using forward and backward models to construct MDP homomorphism is interesting, but no details discussion on when is it possible to learn these forward and backward models, and the numerical experiments may not be enough in demonstrating the benefits of the proposed method.

---

> ### Author Response · Authors · 2022-11-16
> **response to MMzC (1/2)**
>
> Thank you for taking the time to consider our work and for raising issues that, when addressed, will improve the overall manuscript. We respond to specific concerns below.
>
> **Regarding comments about stochastic environments**
>
> Thank you for highlighting your concerns about stochastic environments. We have made the following improvements.
>
>
> (1) We have performed extra experiments with the well known sticky-actions protocol (Machado et al. 2018) on Asterix to be added to the manuscript shortly.
>
> (2) We have added clarity to the manuscript about the predator prey environment, namely *we highlight that it is in fact stochastic*
>
> (3) We have added extra descriptions about the limitations of our approach in stochastic environments and how to deal with them, which we quote below:
>
>
> “A limitation of our method is that we have assumed that for every state-action pair, at least one equivalent canonical state-action can be computed. Given a model of the environment, this is often true but it is not guaranteed for a variety of reasons. For example, near borders in a grid world there is no way to travel left to a border state that has a border on its right. We only found this to significantly affect results in the tabular setting, which we mitigate by reverting to vanilla Q-learning table when necessary (that is learned in parallel).
>
> Furthermore, stochasticity could lead to issues with the current formulation of our approach. If a transition is multimodal and the models predict the mean of the two modes then the value network will be presented with states that do not actually exist in the environment \citep{hafner2019learning}. In Section \ref{limitations}, we discuss how recent developments in recurrent state space models could sample from a distribution of predicted transitions in future work \citep{hafner2019learning}. Lastly, even if forward dynamics are deterministic the previous state could be unpredictable as there may not be a unique previous state for a given previous action---again the model based reinforcement learning provides some ideas for addressing this in future work \citep{yu2021learning}.”
>
> **Regarding ambiguous backward predictions**
>
> Thank you for bringing up this concern, we have added your concerns to the manuscript so that the reader can better understand the limitations. This can be found in the quoted text above that has been added to the manuscript. This issue has been well studied by Yu et al. 2021 which could inform future improvements to our algorithm. In practice, future work could make backwards predictions with probabilistic models and if the model predictions have high uncertainty revert to vanilla Q-learning (as we did for tabular Q-learning).
>
> **Limitations discussion**
>
> The following statement about discussions of limitations is not true:
>
> "there may be conditions under which the described forward and backward models exist, but there are no discussions about in which cases is it possible to learn the forward and backward models in the paper."
>
> We agree with you that this is important to discuss. We included a discussion in our original manuscript---please see the following excerpt from Section 6:
>
> "What’s more, despite recent progress in model based learning in high dimensional environments Hafner et al. (2019; 2020); Saxena et al. (2021); Ye et al. (2021), a dynamics model can be difficult to learn for a number of reasons. Firstly, stochastic dynamics can make environment transitions multimodal, meaning point predictions of future or previous states are not adequate. In theory, this can be dealt with using stochastic models Hafner et al. (2019) of forwards and backwards dynamics that could be sampled when abstracting state-action pairs. Even worse, dynamics can be completely unpredictable Kendall & Gal (2017); Mavor-Parker et al. (2021), in which case aleatoric uncertainty predictions could be used to signal that equivalent effect abstraction should temporarily revert to vanilla Q-learning. Furthermore, backwards transitions could also be impossible to predict because an equivalent state does not exist—as described in Section 1. In this case, uncertainty predictions could also be used to avoid using mappings that are out of distribution with the model training data (because they are not possible) Kendall & Gal (2017). Integrating the described probabilistic transition models into equivalent effect abstraction is an interesting direction for future work."

---

> > ### Author Response · Authors · 2022-11-16
> > **response to MMzC (2/2)**
> >
> > **Regarding analytical/empirical statements about the reduced MDP**
> >
> > We appreciate the request for analytical/empirical analysis of how much smaller the state-action space is. We did state the following in the original manuscript:
> >
> > "By finding an equivalent canonical state-action pair for each experienced state-action pair we reduce the number of state-action pairs that an off-policy algorithm must learn to estimate—effectively reducing the size of the space of values from S × A to S × 1."
> >
> > To be more precise, we have the updated the statement to now read:
> >
> > “...effectively reducing the size of the space of values from S × A to S × 1 + N, where $N \in \mathcal{S} \times \mathcal{A}$ are the states where the canonical action is not computable because of the edge cases described in Section 2 (e.g. stochastic environments, no unique backwards action or transitions that do not exist in the environment such as traveling left to a state with a border to its right).”
> >
> > Nevertheless, we see that adding specific analysis (especially for the tabular environment) provides context. We have addressed this in our updated manuscript with the following statements.
> >
> > “In the gridworld environment, we reduce the size of the state action space and hence the number of Q-values (when going left) from $216$ to $67$. In this case, the number of states in the reduced action space is not reduced by the cardinality of the action space exactly. This is because of the 13 states where it is not possible to reach them by traveling left because there is a border to the right.”
> >
> > and
> >
> > “In the cartpole task, we did not find it necessary to avoid any canonical state action pairs that are not computable---meaning the reduction of the size of the state action space from $\mathcal{S}\times\mathcal{A}$ to $\mathcal{S}\times\mathcal{1}$.”
> >
> > and
> >
> > “In Asterix it is always possible to reach every state by doing the canonical action---meaning we reduce the size of the state-action space by a factor of 6 (the number of actions in the action space).”
> >
> > **Updating clarity about the predator prey claims**
> >
> > Thank you for raising concerns about the claims surrounding the predator prey environments. These claims were not as clear as they could have been and we have updated them accordingly in the manuscript. We have changed the text to now read:
> >
> > “If a model can be obtained before online policy learning, then equivalent effect abstraction can deliver an improvement over vanilla DQN on the stochastic predator prey environment. However, for this environment, the number of episodes to learn a policy is extremely small (~5), so learning a model of the environment online is not practical. As a policy can be learned very quickly, it is not a particularly useful environment to test sample efficiency---it instead serves as a useful benchmark (as demonstrated in \cite{van2020mdp}) to address the possible limitations of MDP homomorphisms in stochastic environments.”
> >
> > **Assumption of pretrained model**
> >
> > Thank you for your concerns about the lack of model training in Minatar Asterix and the gridworld experiment. In gridworld, we do compare to Q-planning, which is the natural planning algorithm baseline to compare to. For Asterix, is there a particular baseline you had in mind that should be compared to?
> >
> > **Regarding the summary of the review**
> >
> > Part of the summary of the review reads:
> >
> > “The idea of using forward and backward models to construct MDP homomorphism is interesting, but no details discussion on when is it possible to learn these forward and backward models,”
> >
> > For clarity, we would again like to point out that we do discuss when model learning is not possible. Please see our response above and Section 6 in the original manuscript.

---

> ### Author Response · Authors · 2022-11-25
> **updates to manuscript MMzC**
>
> Thank you again for suggesting ways to improve our paper and to further prope the applicability of our algorithm in stochastic environments.
>
> In light of our updates to the discussion of the limitations, is there anything else you think should be adressed? Furthermore, are our updates (experiments in stochastic environments and analysis of the reduction of the state action space) based on your main concerns satisfactory?

---

### Official Review · Reviewer_eZfk · 2022-10-28

**Confidence:** 4
**Correctness:** 3
**Technical Novelty And Significance:** 3
**Empirical Novelty And Significance:** 3
**Recommendation:** 3

**Clarity, Quality, Novelty And Reproducibility:**

Clarity:

1. Clarity is needed on whether it is $\sigma(s)$ or $\sigma(s, a)$ in Equation 6. The right-hand side has $a$ but left-hand side has $s$. The main text says $\sigma(s)$ but Algorithm 1 uses $\sigma(s, a)$. Tying the examples presented on the first two page to Equation 6 will help a lot.

2. What is the idea of canonical action?

3. In Algorithm 1, when will the condition $\bar{s} = \sigma(s, a) \in S$ not be true? Doesn't $\sigma$ always output a value in $S$? And how does this generalize to function approximation setting?

4. The GreedyPolicy takes $s$ as input but the function definition of GreedyPolicy takes $s$ and $a$ as input.

Presentation

1. Consider moving a short pseudocode, particularly, for function approximation to the main paper.
2. Why do we set $s_t = s_{t+1}$ in Alg 1, when there is a for loop over $t$.

**Strength And Weaknesses:**

Strength:

1. State abstraction is an important topic
2. This paper presents experiments on both tabular and non-tabular settings.

Weakness:

1. It is unclear what is being learned by $\sigma(s)$? Does it satisfy Equation 3? What is the intuitive idea of $\bar{a}$? Why is it chosen arbitrarily? What theoretical properties does $\sigma(s)$ satisfy? There appears to be a conceptual gap in the motivations in the first two pages, and the way $\sigma$ is extracted from combining the forward and backward model. I hope the discussion period will help make this clearer.

Further, the paper currently lists handwavy statements such as:
Figure 2 caption, _"we take advantage of the fact that state-action pairs that lead to the same state usually have equivalent values."_
and the paragraph before Section 4 says _"Lastly, we have assumed that for every state-action pair at least one equivalent canonical state-action exists, which is often true but not guaranteed"_

Can these conditions be expressed formally? and in what realistic settings are these conditions satisfied?

2. The proposed objective performs generative modeling which requires more samples to learn accurately. A more sample-efficient approach would be to do multi-class classification over a small label space, as it done by action-prediction or contrastive learning-style approaches.

3. The paper ignores related work on learning state abstraction with function approximation including autoencoders, contrastive-learning approaches. A few examples are listed below, but there are many more:

- (Autoencoder) #Exploration: A Study of Count-Based Exploration for Deep Reinforcement Learning, Tang et al. NeurIPS 2017
- (Contrastive Learning) Kinematic state abstraction and provably efficient rich-observation reinforcement learning, Misra et al., ICML 2020
- (Generative Modeling) FLAMBE: Structural Complexity and Representation Learning of Low-Rank MDPs, Agarwal et al., NeurIPS 2020
- (Action prediction) Planning from Pixels using Inverse Dynamics Models, Paster et al., ICLR 2021

The paper also misses related work that learn mapping between a given MDP and an abstract MDP. For example, see

- Provably Efficient Model-based Policy Adaptation, Song et al., ICML 2020
- Provably Sample-Efficient RL with Side Information about Latent Dynamics, Liu et al, NeurIPS 2022

While the proposed approach is new, it will nevertheless benefit readers to situate the work in the literature.

4. Algorithm 1 is very hard to understand. See clarity below.

**Summary Of The Paper:**

This paper attempts to learn a state abstraction based on MDP homomorphism. The goal is to reduce the size of state-action space and allow for faster RL. The proposed approach learns a backward model $(B)$ and a forward model $(F)$. The forward model $F(s, a)$ is trained to predict the next state given the current state $s$ and action $a$. The backward model $B(s', a)$ is trained to predict the previous state given the current state $s'$ and last action $a$. The state abstraction mechanism maps a state (or a state-action?) pair to a value $\sigma(s)$ such that:

$\sigma(s) = B(F(s, a), \bar{a})$,

where $\bar{a}$ is an action called canonical action. Algorithm 1 shows that this action is chosen arbitrarily.  Using this approach, one can do Q-learning on $\sigma(s), \bar{a}$ values instead of $s, a$. A series of experiments on tabular and function approximation settings are presented that show promise of this approach.

**Summary Of The Review:**

Lack of clarity is the main reason for my current score of reject. Based on author response and discussion period, I'll consider changing my score. In particular, clarity on what $\sigma$ is learning will be most helpful.

---

> ### Author Response · Authors · 2022-11-19
> **Response to eZfK (1/2)**
>
> We thank reviewer eZfk for taking the time to review our work in detail. We especially appreciate that they have provided specific instructions on how to improve the clarity of the paper. After reading the positive reviews on the clarity of our writing (e.g. Reviewer MMzC found “The paper is generally well written and easy to follow” and DLp5 described our paper as “overall well written and easy to understand”) we were disheartened to see eZfK found it unclear---we hope that the changes we have made based on your constructive suggestions can sway your opinion on our work’s clarity so that a consensus can be reached with the other reviewers. We describe the changes we have made below.
>
> **We have updated the text to address the following questions and improve clarity:**
>
> **(1) “It is unclear what is being learned by \sigma? Does it satisfy Equation 3?”**
>
> To make this more clear we have added the following text after Equation 6:
>
> “We describe the combination of the forwards and backwards model as $\sigma$ solely to be consistent with the previous literature \cite{van2020mdp} (i.e. it is a state homomorphism). In simple terms, $\sigma$ is just a mapping to another state-action pair that leads to the same next state. To construct $\sigma$, we use a combination of a forwards model prediction and a backwards model prediction. The first model is conditioned on the action $a$ you would like to know the value for and the reverse action $\bar{a}$ is the canonical action set by the practitioner.”
>
> **(2) “What theoretical properties does \sigma satisfy? Does it satisfy Equation 3?”**
>
> We have added the following text to describe the theoretical properties of \sigma more precisely.
>
> “It is important to stress that with the described forwards and backwards models  we have obtained a mapping $\sigma(\mathbf{s}, \bar{a})$, which satisfies Equation 3. This mapping $\sigma(\mathbf{s}, \bar{a})$ can take any state-action pair and map it into a smaller state-action space---matching it with a state-action pair that has equivalent value \citep{ravindran2001symmetries}. As our mapping satisfies Equation 3, it also assumes the MDP homomorphism properties developed by \citep{ravindran2001symmetries}, meaning the policy we learn with our canonical state action pairs can be ``lifted'' \citep{van2020mdp} and used as a policy in the experienced environment.”
>
> **(3) “What is the intuitive idea of $\bar{a}$? Why is it chosen arbitrarily?”**
>
> To give the reader a better understanding of what $\bar{a}$ is we have added the following text to Section 3.
>
> “In theory, one could calculate multiple equivalent state-action pairs for a given state-action pair---by querying the backwards model with multiple different actions. However, we are trying to reduce the overall state-action space that our policy learns in, so instead we select one action to be our ''canonical'' action and all other state-action pairs are mapped into its ``canonical'' reference frame. In Figure 2, the canonical action is moving right---meaning our Q-network only has to learn about moving right and we can map all right values to their equivalent left values with our forwards and backwards models. The setting of the canonical action is essentially a hyperparameter. In some environments such as Cartpole, the canonical action can be selected randomly. However, in other environments backwards state predictions are not always possible for a given action in a given state. In these environments, canonical actions need to be selected more carefully. Next, we describe the aforementioned situations where backwards predictions cannot be made.”
>
> **Regarding “hand wavy” statements**
>
> We did not intend for the statements to be hand wavy. We have updated the quoted sentences to be more rigorous:
>
> "We take advantage of the fact that state-action pairs that lead to the same state usually have equivalent values."
>
> Has been updated to:
>
> "We take advantage of the fact that, if rewards only depend only on states (which is true for most RL benchmark tasks), then state-action pairs that lead to the same state have equivalent values."
>
> To improve the clarity of paragraph before section 4, it has been updated to the following:
>
> “Furthermore, stochasticity could lead to issues with the current formulation of our approach. If a transition is multimodal and the models predict the mean of the two modes then the value network will be presented with states that do not actually exist in the environment. In Section \ref{limitations}, we discuss how recent developments in recurrent state space models could be used to sample from a distribution of predicted transitions in future work \citep{hafner2019learning}. Lastly, even if forward dynamics are deterministic the previous state could be unpredictable as there may not be a uniqe previous state for a given previous action---again the model based reinforcement learning provides some ideas for addressing this in future work \citep{yu2021learning}.”

---

> > ### Author Response · Authors · 2022-11-19
> > **response to eZfK (2/2)**
> >
> > **Regarding gaps in literature review**
> >
> > Thank you for your suggestion to add more representation learning papers to the literature review. Representation learning and abstraction in RL is a very large scope, so we did our best to reference the papers most relevant to MDP homomorphisms. Nevertheless, we found your literature suggestions very useful and have added the following to the main text:
> >
> > “\citep{misra2020kinematic} use contrastive learning to learn a similar abstraction---grouping together state-actions that will pass through or have passed through same state”
> >
> > And
> >
> > “A related exploration method shows simple autoencoders can learn binary abstract representations useful for logging of what states an agent has experienced (Tang et al., 2017).”
> >
> > **Conditioning the homomorphism on actions**
> >
> > We agree that how the equations were previously written is confusing, as it is not explicitly clear whether the mapping is conditioned on actions. To clear up any confusion, the backwards model is learned for only one action (the canonical action) and as the mapping learned depends on which action is chosen to be canonical. We have updated the equations accordingly. Equation 6 now reads:
> >
> > \bar{\mathbf{s}} = \sigma(\mathbf{s}, \bar{a}) =  \mathcal{B}_{\phi}(\mathcal{F}_{\theta}(\mathbf{s}, a), \bar{a})
> >
> > **Detailed explanation of when is model learning possible**
> >
> > Based on your comments we have added detailed comments about when learning our homomorphism is possible.
> >
> > “A limitation of our method is that we have assumed that for every state-action pair, at least one equivalent canonical state-action can be computed. Given a competent model of the environment, this is often true but it is not guaranteed for a variety of reasons. For example, near borders in a grid world there is no way to travel left to a border state that has a border on its right. We only found this to significantly effect results in the a tabular setting, which we mitigate by reverting to vanilla Q-learning table when necessary (that is learned in parallel).”
> >
> > This paragraph is followed by discussions around stochasticity quoted earlier.
> >
> > **Clarity for algorithm one**
> >
> > Thank you for pointing out issues in the clarity of algorithm 1. We have significantly updated its format to improve its clarity. Please see the updated algorithm in the new manuscript.
> >
> > **In response to “Tying the examples presented on the first two page to Equation 6 will help a lot.”**
> >
> > We now explicitly reference this example when describing canonical actions. (see response above).
> >
> > **Regarding Algorithm 1 and suggestions for pseudocode.**
> >
> > We apologize that Algorithm 1 seems to have just added confusion when its intended purpose was to add clarity. Based on your comments we have scraped Algorithm 1 in its previous form and rewritten it. It now reads as pytorch pseudo code that can be used in the forward pass of pytorch style Q-network. Equipped with a forwards and backwards model, these are all the changes to a DQN implementation that are required.

---

> ### Author Response · Authors · 2022-11-25
> **updates to manuscript eZfK**
>
> Thank you again for helping us understand how to make the presentation of our algorithm more clear. In the updated manuscript, have we adressed your concerns about the clarity of the text?

---

### Official Review · Reviewer_DLp5 · 2022-11-04

**Confidence:** 4
**Correctness:** 4
**Technical Novelty And Significance:** 3
**Empirical Novelty And Significance:** 3
**Recommendation:** 6

**Clarity, Quality, Novelty And Reproducibility:**

The paper is overall well written and easy to understand. There are, however, a few smaller issues. The first is that it is unclear why both the Algorithm I and the results for the Minatar experiments have been moved into the appendix while their description and the reference to them are in the main text. In particular the absence of the algorithm in the main text near the reference (without explicitly stating that it is in the Appendix) may cause the reader to pause. It would be better if both of these items would be moved in line with the text and the descriptions to make it easier for the reader.
The approach is novel and while somewhat limited in its application in the current form (see comments above), it provides an interesting direction for on-line increase of sample efficiency in RL.
The paper does provide a good description of the experimental settings and listing of the hyper parameters in the Appendix make it easier to reproduce the results.

**Strength And Weaknesses:**

The paper presents an approach to constructing a homomorphic abstraction by identifying equivalent effect state-action pairs and using these to map to a canonical element, thus reducing the complexity of the value function and thus potentially accelerating learning of policies on the abstract representations. The strengths of this approach lie in the ability to derive the abstraction form a set of environmental experiences without the need for a prior model. In addition, the presented model only requires one step equivalences for its construction, making it more applicable for on-line learning settings. However, the significance of this is somewhat limited by the prior assumptions on the task, namely that rewards are purely a function of the state (making it difficult to account for variable action costs, except by embedding them in the state through something like a remaining/consumed energy value), the need for deterministic transitions, and the assumption that an inverse model exists in most situations (i.e. that in most situations a state can be reached using every action), which seems to potentially significantly limit the application domain of the method.
While the authors include a discussion of some of the limitations at the end of the paper, it would be useful if some more general discussion could be included earlier on that detailed the effect of these limitations and in particular what it implies for the types of problems that the method is applicable to. Also, it might be worth discussing and considering whether some of the assumptions can be softened to extend the applicability of the approach.

Another limitation of the homomorphism approach in this paper compared to the more general homomorphism framework seems to be that it can not easily abstract in situations where different regions of the state space have similar dynamics, i.e. where different task instances exist within the overall state space. This is an important property to extract if the learning agent might be placed into different versions of an environment and asked to solve the same task as shown in previous Homomorphism work for policy generalization such as [Ravindran, Balaraman, and Andrew G. Barto. "Model minimization in hierarchical reinforcement learning." International Symposium on Abstraction, Reformulation, and Approximation. Springer, Berlin, Heidelberg, 2002.] of [Rajendran, Srividhya, and Manfred Huber. "Learning to generalize and reuse skills using approximate partial policy homomorphisms." 2009 IEEE International Conference on Systems, Man and Cybernetics. IEEE, 2009.]. While alluded to in the limitations section, a further discussion of how the presented approach might be combined with other homomorphism concepts would be useful to the reader to assess the full potential of the technique.

There is a typo in the first paragraph of Section 4.3: ""...an agent much chase..." should be "...an agent must chase..."

**Summary Of The Paper:**

This paper presents a novel approach to abstraction in Reinforcement Learning in discrete action spaces by construction a homomorphic representation through equivalent effect state-action pairs. The approach constructs forward and backward models which allow it to reduce the complexity of the value function representation by mapping equivalent state-action sets to canonical entries. The aim is to increase sample efficiency in subsequent learning tasks and the potential benefits are demonstrated in a number of standard experiments in maze and simple control environments.
The main contributions of the paper are the introduction of the new approach which allows the construction of a homomorphic abstraction (although a somewhat limited one) largely on-line from simple transition experiences without prior knowledge of the complete transition dynamics. While somewhat limited in its applicability due to its limitation to deterministic transition tasks, it introduces an interesting concept and illustrates its potential benefits in a number of experiment domains.

**Summary Of The Review:**

The paper presents a novel approach to learn a simple (and somewhat limited) homomorphic representation for a problem using equivalent effect state-action pairs in order to reduce small efficiency in RL. This approach allows the abstraction to be learned efficiently on-line form a small number of transition experiences, making it easily applicable to some practical problems. While the underlying assumptions on the problem (reward only depends on state, deterministic transitions, existence of an inverse transition function in most states) somewhat limits the applicability (and thus the significance), the approach points out a potentially useful direction for abstraction work and could thus lead to beneficial follow-up work. This make it, in my view, of sufficient interest to the community to warrant publication.
The experiments in the paper illustrate the potential benefit of the technique.
The paper also includes a discussion of limitations and potential extensions. This discussion should be slightly further extended (and part of it moved earlier in the paper) to make sure that readers can fully understand the limitations of the presents part but also be able to assess possible extensions in the future to mitigate these limitations.

---

> ### Author Response · Authors · 2022-11-16
> **Response to DLp5**
>
> We thank reviewer DLp5 for their careful reviewing of our paper. We also appreciate that they believe that "the approach points out a potentially useful direction for abstraction work and could thus lead to beneficial follow-up work. This makes it, in my view, of sufficient interest to the community to warrant publication." Beyond these positive comments, DLp5 has provided some excellent suggestions on how to improve the manuscript, which we address below.
>
> **Regarding task assumptions**
>
> Thank you for raising your concerns with how we have discussed the limitations of our approach in different environments. To make this discussion more verbose, we have followed your advice and added the following to Section 3 of the paper.
>
> “A limitation of our method is that we have assumed that for every state-action pair, at least one equivalent canonical state-action can be computed. Given a model of the environment, this is often true but it is not guaranteed for a variety of reasons. For example, near borders in a grid world there is no way to travel left to a border state that has a border on its right. We only found this to significantly affect results in the tabular setting, which we mitigate by reverting to vanilla Q-learning table when necessary (that is learned in parallel).
>
> Furthermore, stochasticity could lead to issues with the current formulation of our approach. If a transition is multimodal and the models predict the mean of the two modes then the value network will be presented with states that do not actually exist in the environment \citep{hafner2019learning}. In Section \ref{limitations}, we discuss how recent developments in recurrent state space models could be used to sample from a distribution of predicted transitions in future work \citep{hafner2019learning}. Lastly, even if forward dynamics are deterministic the previous state could be unpredictable as there may not be a unique previous state for a given previous action---again the model based reinforcement learning provides some ideas for addressing this in future work \citep{yu2021learning}.”
>
> While we agree that it is a limitation of our approach that it only defines rewards based on successor states, we do not believe that this severely limits the applicability of our method. For example, gym control environments, atari games and mujoco environments define reward in terms of states. Could you suggest some environments that don't define rewards in this way so that we can address them in the limitations section?
>
> **Regarding environment stochasticity**
>
> We agree that requiring deterministic transitions would be a limiting aspect of our work. Firstly, we would like to point out that the predator prey environment is in fact stochastic (it is impossible to predict where the prey will move next). Furthermore, we will be adding more results with sticky actions (Machado etal. 2018) to the updated manuscript.
>
> **Regarding previous work on symmetry based homomorphisms**
>
> It is true that our work does not exploit symmetries that seminal works on MDP homomorphisms use to their advantage. We have added the following to the conclusion section to explain more about how our approach could be combined with symmetry based approaches for even greater sample efficiency.
>
> “Additionally, equivalent effect abstraction could also be integrated into existing homomorphic MDP methods that rely on symmetries to further reduce the size of the abstract state-action space \cite{van2020mdp}. In practice, this would mean using a homomorphic Q-network \cite{van2020mdp} but modified to only have one action output as shown in Algorithm 1. Alternatively, planning could be done with an abstract homomorphic MDP \cite{van2020plannable}, which is then reduced further with equivalent effect abstraction.”
>
> We do not see this as a limitation of our approach---we are aiming to reduce MDPs by exploiting a separate kind of structure that, importantly, can be learned from experience. As far as we are aware, there is no reliable way to learn environment symmetries from experience.
>
> **Regarding formatting**
>
> Thank you for pointing out that placing the algorithm in the appendix makes life difficult for the reader. We have reformatted the algorithm into pytorch style pseudo code for our value network that we hope adds clarity. We have also now put this in the main body. Thank you also for pointing out the typo that we have now addressed.
>
> **Limitations and discussion**
>
> As per your suggestion we have moved parts of the limitations earlier in the paper and also extended the discussion section. See response quoted above and also our updated manuscript to be uploaded in the next few days.

---

> ### Author Response · Authors · 2022-11-25
> **Updates to manuscript DLp5**
>
> We again would like to thank DLp5 for their helpful suggestions, in particular for suggesting we increase the discussion of limitations. Can you confirm whether these changes adequately adress your concerns about the limitations of our algorithm.

---

### Author Response · Authors · 2022-11-15
**Rebuttals Update**

We would like to thank all the reviewers for their valuable feedback. We will respond to each reviewer individually over the coming days and then upload an updated manuscript that aims to address reviewer concerns before the Friday deadline.

---

### Author Response · Authors · 2022-11-19
**Updates to manuscript**

We thank the reviewer's for their valuable comments that have helped us to improve the manuscript. We have uploaded the new manuscript with changes marked in blue for easy comparison to our previous draft. We have made many changes based on your suggestions, but the main points to consider are:

(1) sticky actions experiments for asterix to test our method in stochatic environments

(2) rewriting algorithm in simple pytorch pseudo code for better clarity

(3) more rigorous discussion of learned mappings are possible and also more clear discussion of the introduction of our MDP homomorphism

If you find the manuscript changes in blue distracting, you can find the previous manuscript version (without the changes marked) in the rebuttal revision proceeding the current version.

---

### Decision · Program_Chairs · 2023-01-20

**Decision:**

Reject

**Justification For Why Not Higher Score:**

The method is based on very restrictive assumptions. The scores are quite low.

**Justification For Why Not Lower Score:**

N/A

**Metareview: Summary, Strengths And Weaknesses:**

The reviewers have brought up issues regarding the restriction of assumptions, the clarity of the paper, as well as several other issues. The authors have significantly revised the paper to answer the reviewers' comments. Although some of the concerns are addressed or explained, a more careful reevaluation by several reviewers is needed.

At the end, we have two positive reviews with the score of 6 and two negative reviews with score of 3.

Given the current scores, the volume of revisions in the paper, and the restriction of the assumptions and the setting when the method works, I believe the paper is not suitable for publication at the moment.